# Correlation between Psychosomatic Assessment, Heart Rate Variability, and Refractory GERD: A Prospective Study in Patients with Acid Reflux Esophagitis

**DOI:** 10.3390/life13091862

**Published:** 2023-09-03

**Authors:** Hsin-Ming Wang, Pao-Yuan Huang, Shih-Cheng Yang, Ming-Kung Wu, Wei-Chen Tai, Chih-Hung Chen, Chih-Chien Yao, Lung-Sheng Lu, Seng-Kee Chuah, Yu-Chi Lee, Chih-Ming Liang

**Affiliations:** 1Division of Hepato-Gastroenterology, Department of Internal Medicine, Kaohsiung Chang Gung Memorial Hospital and Chang Gung University College of Medicine, 123 Ta Pei Road, Niao-Sung Dist., Kaohsiung 833, Taiwan; ambulance1027@gmail.com (H.-M.W.); paoyuan813@gmail.com (P.-Y.H.); d5637700@cgmh.org.tw (S.-C.Y.); luketai1019@gmail.com (W.-C.T.); totoro631105@yahoo.com.tw (C.-H.C.); chihchienyao@gmail.com (C.-C.Y.); u501118@gmail.com (L.-S.L.); sengkeechuah@gmail.com (S.-K.C.); 2Department of Psychiatry, Kaohsiung Chang Gung Memorial Hospital and Chang Gung University College of Medicine, Kaohsiung 833, Taiwan; mingkung180@gmail.com

**Keywords:** gastroesophageal reflux disease (GERD), psychosomatic assessments, heart rate variability (HRV), refractory GERD, personalized treatment strategies

## Abstract

Background: Gastroesophageal reflux disease (GERD) affects a significant proportion of individuals, with life stress being a contributing factor. This study aimed to investigate the correlation between psychosomatic evaluations, heart rate variability (HRV), and GERD in a cohort of individuals. Additionally, the study aimed to analyze the sequencing changes following proton pump inhibitor (PPI) treatment and identify predictive factors associated with refractory GERD. Methods: A prospective cohort of 105 individuals with reflux esophagitis and a control group of 50 participants without acid reflux symptoms were enrolled. Psychosomatic evaluations, including GERDQ, GERDQLQ, RSI, BAI, BDI, and SSS-8, were assessed at baseline and during treatment. HRV parameters were also evaluated. Multivariate analysis was used to identify predictive factors for refractory GERD. PPIs were administered regularly for the initial 2 months and then used on-demand. Refractory GERD was defined as less than 50% improvement in symptom relief or GERDQLQ score ≥ 20 after 8 weeks of PPI treatment. Results: The GERD group had higher scores in all psychosomatic evaluations compared to the control group (all *p*-values < 0.001). There were no significant changes in any parameters of HRV before and after treatment in the GERD group. Strong and consistent correlations were observed between GERD symptoms and psychological scores (BAI, BDI, and SSS-8) across all time points (W0, W4, and W8). Sequential reductions in GERD symptom scores and psychosomatic evaluations were observed during the initial eight weeks of treatment. Higher GERDQ (≥10) and SSS-8 (≥12) scores were predictive of refractory GERD (*p* = 0.004 and *p* = 0.009, respectively). Conclusions: This study emphasizes the importance of considering physiological and psychological factors in the management of GERD. Psychosomatic evaluations provide valuable insights for assessing and treating GERD patients. Integrating stress management and comprehensive assessments into personalized treatment strategies is crucial.

## 1. Introduction

The prevalence of gastroesophageal reflux disease (GERD) has significantly increased in recent decades [1]. GERD has a negative impact on patients’ quality of life (QoL) and daily activities [2]. Proton pump inhibitors (PPIs) are commonly used as first-line therapy for GERD; however, a substantial proportion of patients do not respond adequately to PPI treatment [3,4,5]. Failure of PPI treatment to resolve GERD-related symptoms has become the most common presentation of GERD in clinical gastroenterology [6]. Persistent GERD symptoms are reported in 17–32% of primary care patients receiving PPI therapy and 45% of participants in observational primary care and community-based studies [7]. Functional heartburn and reflux hypersensitivity account for over 90% of heartburn patients who failed twice-daily PPI treatment [8,9]. Psychological factors, including anxiety and depression, are also prevalent in patients with GERD, and higher psychological scores for neuroticism, anxiety, and depression are positively correlated with heartburn symptoms [10,11]. However, objective tools to measure the psychosomatic level in GERD populations are currently lacking. Neuroimaging research has suggested a potential association between heart rate variability (HRV) and cortical regions, such as the ventromedial prefrontal cortex, involved in evaluating stressful situations. A meta-analysis study demonstrated that HRV is influenced by stress and can be used to objectively assess psychological well-being and stress levels [12]. Altered HRV, indicating severe autonomic dysfunction, has been found in GERD patients compared to controls [13]. Understanding the sequential patterns of HRV can help identify patients with anxious depression who may benefit from antidepressant medications or alternative treatment approaches [14]. Based on the available evidence, the objective of this study was to investigate the correlation between different psychosomatic evaluations and heart rate variability (HRV) among individuals diagnosed endoscopically with reflux esophagitis. Additionally, the study sought to analyze the alterations in sequencing following PPI treatment and identify potential predictive factors associated with refractory GERD.

## 2. Patients and Methods

### 2.1. Ethical Requirements

The study protocol was approved by the Institutional Review Board and the Ethics Committee of Kaohsiung Chang Gung Memorial Hospital in Taiwan (permitted number 202000210B0). We obtained written informed consent from all enrolled patients before participation. 

### 2.2. Study Cohort, Inclusion and Exclusion Criteria 

For this study, a total of 105 individuals presenting with acid reflux symptoms and diagnosed with erosive esophagitis through endoscopic examination were prospectively recruited from the outpatient department between 1 January 2020 and 30 November 2022. Additionally, a control group consisting of 50 age and sex-matched volunteers was included for comparative analysis. The primary criteria for inclusion were the absence of typical reflux symptoms like heartburn and regurgitation. Only one participant of the control group reported a recent history of epigastric pain. As for endoscopy, controls did not routinely undergo endoscopy unless their medical history indicated a necessity. The objective was to ascertain the absence of erosive esophagitis (EE), and any participants identified with EE were subsequently excluded. The general exclusion criteria applied to both patient and control groups, which comprised individuals with the following conditions: alcohol abuse, recent non-steroid anti-inflammatory drugs (NSAIDs) exposure, and non-adherence to recommended lifestyle modifications; prior upper gastrointestinal surgery; comorbidities such as seizure, scleroderma, autonomic or peripheral neuropathy, or myopathy; and an inability or unwillingness to provide informed consent. Pregnant or lactating women, and those planning to conceive, were also excluded. Patients with major depression and a history of suicide attempts were not considered and were, instead, referred to a psychologist for further management.

We assessed the Chinese gastroesophageal reflux disease questionnaire (GERDQ), the gastroesophageal reflux disease–quality of life questionnaire (GERDQLQ), the reflux symptom index (RSI), the psychosomatic assessment (Beck Anxiety Inventory (BAI), the Beck Depression Inventory (BDI), and the Somatic Symptom Scale-8 (SSS-8) at baseline and after treatment at weeks 4 and 8. HRV was also evaluated at weeks 0 and 8. Prior to initiating proton pump inhibitor (PPI) treatment, the patients were provided with comprehensive education regarding lifestyle modifications and dietary adjustments by the healthcare professionals. The PPI therapy duration: Two months of once daily dexilansoprazole 60 mg for grade A or B erosive esophagitis, followed by on-demand use if GERD symptoms recurred. The PPI was titrated to twice-daily PPI in the 2nd month for the patients who exhibited incomplete or partial response to the standard PPI regimen (once daily) [15,16,17,18]. At the end of initial treatment (8 weeks), all patients were switched to an on-demand therapy, using 60 mg dexlansoprazole successively for 3 days if the GERD symptom relapsed [19]. “Refractory GERD” is classified as exhibiting less than a 50% improvement in symptom relief and life quality (as measured by the GERDQLQ), or achieving a score equal to or greater than 20, in response to treatment with PPIs [15,20]. Refractory GERD patients were advised to undergo further esophageal manometry and 24 h pH monitoring, as per clinical guidelines, if patients agreed to it [21]. 

### 2.3. Acquisition and Analysis of Heart Rate Variability Data

Data pertaining to Heart Rate Variability (HRV) were captured from each participating patient and control individual in a state of short-term rest, utilizing an HRV monitor (LR8Z11) provided by Yangyin Corp., Taipei, Taiwan [22]. The electrocardiographic (ECG) information was collected at a sampling rate of 512 Hz, with a working voltage of 5 V, a band-pass filter operating within the range of 0.05–40 Hz, and a duration of 5 min. These data were subsequently digitized using an embedded analog-to-digital converter (single-channel, 10-bit, 1000 points per second) within the HRV monitor and then transferred to a microcomputer for advanced power spectral analysis of HRV. The procedure of HRV measurement was conducted twice over 5 min in the outpatient department at W0 and W8, prior to any other procedures or interventions, ensuring that the readings were taken when the patient was in a relatively relaxed and undisturbed state after a 10 min quiet sitting period. A comprehensive description and representation of the HRV parameters are provided in Appendix A [23,24,25,26].

### 2.4. Acid Reflux Symptoms Assessments

The gastroesophageal reflux disease questionnaire (GERDQ) consists of six questions and employs two types of Likert scales to evaluate positive and negative predictors. Questions 1, 2, 5, and 6 utilize a standard Likert scale ranging from 0 to 3, while questions 3 (pain in the middle of the upper stomach area) and 4 (nausea) use a reverse Likert scale ranging from 3 to 0. The total GerdQ score can range from 0 to 18, and a score of 8 or higher is generally recommended as the threshold for diagnosing GERD [27]. 

Reflux symptom index (RSI) is a nine-item evaluation instrument designed to measure different symptoms related to Laryngopharyngeal Reflux (LPR). Each of these nine items is rated on a scale that goes from zero (indicating no symptoms) to five (indicating severe symptoms). The highest possible cumulative score is 45, which represents the most severe symptomatology. An RSI score exceeding 13 is typically viewed as abnormal and indicative of LPR [28]. 

### 2.5. Psychosomatic Assessments

The Beck Anxiety Inventory (BAI) constitutes a 21-item, 4-point, self-administered scale constructed to quantify an individual’s self-disclosed anxiety level. This inventory’s score spectrum spans from 0 to 63, adopting a 0–3 scale for responses. Thus, cumulative scores may oscillate between 0 and 63, where 0–7 signifies normal level, 8–15 mild anxiety, 16–25 moderate anxiety, and 26–63 severe anxiety [29]. The Traditional Chinese version of the BAI showcases analogous psychometric characteristics [30]. 

The Beck Depression Inventory-II (BDI-II) is a 1996 modification of the original BDI, which redefined numerous diagnostic parameters for Major Depressive Disorder [31]. The BDI-II comprises around 21 items, each response scoring within a 0 to 3 range. Elevated total scores correspond to more intense depressive manifestations. The standardized thresholds applied diverge from the initial version: 0–13 symbolizes minimal depression, 14–19 mild depression, 20–28 moderate depression, and 29–63 severe depression. The Traditional Chinese iteration of the BDI-II demonstrates similar psychometric attributes [32].

The Somatic Symptom Scale 8 (SSS-8) is a brief self-report questionnaire used to assess somatic symptom burden [33]. The SSS-8 includes the following symptoms: stomach or bowel problems, back pain in the arms/legs/joints, headaches, chest pain/shortness of breath, dizziness, feeling tired/having low energy, and trouble sleeping. Severity categories for the SSS-8 were delineated per scoring ranks, including no to minimal (0–3 points), low (4–7 points), medium (8–11 points), high (12–15 points), and very high (16–32 points) somatic symptom burden. The Chinese adaptation of the SSS-8 possesses adequate reliability and validity, thus supporting its implementation in research and clinical contexts [34].

### 2.6. Characteristics of the Study Populations

A comprehensive set of medical and demographic information was compiled for each patient in the study cohort. This included age, gender, complete medical history, smoking habits, consumption of alcohol, coffee, and tea, dietary practices, notably the ingestion of sweat-inducing and spicy foods [35], and the history of Gastroesophageal Reflux Disease (GERD). Further, we collected specific parameters, including the body mass index (BMI) and observations derived from endoscopy, such as esophagitis classified by the Los Angeles grading system, and the occurrence of hiatal herniation.

### 2.7. Statistical Analysis

Continuous data were presented as means ± standard deviation (SD), and categorical data are presented as frequencies and percentages. One-way analysis of variance (ANOVA) was used for comparing more than two groups. Pearson’s chi-square or Fisher’s exact 2-tailed tests were used for the analysis of categorical data, while continuous variables were analyzed using the *t*-test, where appropriate. An χ^2^ test for linear trends was used to assess the trends of variate scores over time. The Pearson correlation coefficient (r) was employed to assess the linear relationship between GERD symptoms and psychosomatic scores. Two-tailed *p*-values < 0.05 were considered statistically significant. All analyses were performed using the Statistical Package for Social Sciences (SPSS^®^ version 22.0 for Windows, IBM Crop. Armonk, New York, NY, USA).

## 3. Results 

### 3.1. Baseline Characteristics between the GERD and Control Groups

From the initial cohort of 105 patients, seven were unable to be followed-up with, resulting in a total of 98 patients included in the final analysis. Among these patients, 89 (90.8%) had grade A erosive esophagitis and 9 (9.2%) had grade B erosive esophagitis, as determined by endoscopic examination according to the Los Angeles classification system. This group was subsequently compared with a control group composed of 50 individuals. Age appears to be similarly distributed in both groups, with a mean of 50.6 years (±15.1) in the GERD group and 50.2 years (±13.1) in the control group (*p* = 0.854). The variables of sex, BMI, smoking status, alcohol consumption, coffee intake, consumption of betel nut and spicy food, and certain comorbidities such as diabetes, hypertension, coronary artery disease, and chronic kidney disease did not show statistically significant differences between the GERD group and the control group. However, insomnia was significantly more common in the GERD group (56.1% vs. 28.0%, *p* = 0.001). Tea consumption showed a significant difference between the two groups (*p* = 0.023). A total of 62.2% of the GERD group consumed tea, compared to 78.0% in the control group. Regarding symptom scores, the GERD group had significantly higher scores for all indexes, including GERDQ, GERDQLQ, RSI, BAI, BDI, and SSD-8, when compared to control group (all *p*-values < 0.001), which is consistent with their diagnosis and the symptomatic burden of GERD (Table 1).

### 3.2. Sequential Change of Reflux Symptom Score and Psychosomatic Assessments after Treatment

Subsequent to treatment, notable linear reductions were detected in GERDQ, GERDQLQ, RSI scores, and psychosomatic evaluations (BAI, BDI, and SSS-8) during the initial eight weeks (all *p*-values < 0.001) (Figure 1). Nevertheless, the progress associated with GERDQ seemed to reach a plateau at the 16th, 24th, and 48th weeks (Figure 2) (Appendix A).

### 3.3. Correlations between GERD Symptoms and Psychological Scores

Strong and consistent correlations were observed between GERD symptoms and psychological scores (BAI, BDI, and SSS-8) across all time points (W0, W4, and W8). Notably, the strongest correlations involved SSS-8 and GERDQLQ at each time point, with coefficients (r) exceeding 0.7 and *p*-values less than 0.001. These results suggest a significant association between GERD symptoms and psychological factors (Table 2).

### 3.4. Heart Rate Variability Parameters for the Control and the GERD Groups

Table 3 delineates the HRV parameters for the control group and the GERD group, both prior to treatment (W0) and following the completion of treatment (W8). The parameters with significant differences between the control group and the GERD group before the treatment are marked with an asterisk (*). These include SD, TP, VL, HF, and SDNN. The control group showed the lower values than the GERD group before the treatment in the parameters of HRV (28.7 ± 12.6 vs. 39.8 ± 46.3, *p* = 0.035 (SD), 784.0 ± 699.3 vs. 1556.2 ± 3324.4, *p* = 0.030 (TP), 331.7 ± 263.3 vs. 561.1 ± 954.5, *p* = 0.029 (VL), 208.9 ± 318.2 vs. 618.5 ± 1644.6, *p* = 0.030 (HF), and 28.3 ± 12.5 vs. 39.8 ± 46.3, *p* = 0.029 (SDNN)). These differences were found to be statistically significant. However, there were no significant changes in any parameters of HRV before and after treatment in the GERD group. The findings indicated that individuals with GERD display altered regulation of the autonomic nervous system (TP) in comparison to those without GERD. The observed lower values of the HRV parameters in the normal control group suggest potential reductions in sympathetic (VL) and parasympathetic activity (HF) among these individuals. These findings may be indicative of underlying physiological changes associated with GERD.

### 3.5. Predictive Variables for Refractory GERD in Patients with Erosive Esophagitis

Fifteen patients exhibited a refractory response after two months of treatment. In the univariate analysis, the refractory GERD group demonstrated significantly higher scores in GERDQ (10.7 ± 1.8 vs. 8.7 ± 2.3, *p* = 0.002), GERDQLQ (38.5 ± 12.5 vs. 22.9 ± 16.9, *p* = 0.001), RSI (20.9 ± 7.8 vs. 14.1 ± 8.2, *p* = 0.004), BDI (10.4 ± 5.3 vs. 6.5 ± 6.7, *p* = 0.037), and SSS-8 (16.7 ± 7.2 vs. 9.8 ± 6.2, *p* < 0.001) compared to the non-refractory group. However, there was no significant difference observed in HRV parameters between the refractory and non-refractory GERD groups (Table 4). In the multivariate analysis, the predictive factors for refractory GERD were identified as having a GERDQ score ≥ 10 (OR, 95% C.I., 11.028 [2.147–56.654], *p* = 0.004) and an SSS-8 score ≥ 12 (OR, 95% C.I., 7.377 [1.636–33.256], *p* = 0.009), as shown in Table 5. Patients with refractory GERD were advised to undergo esophageal manometry and 24 h pH monitoring. Due to the invasive nature of these procedures, only two patients opted for them. The results revealed ineffective esophageal motility and hypersensitive esophagus. 

## 4. Discussion

Our study provides important insights into the relationship between psychosomatic assessments and HRV in patients with acid reflux esophagitis after PPI treatment. We observed significant differences in HRV and psychosomatic evaluations between the GERD and non-GERD groups, highlighting the relevance of considering both physiological and psychological factors in the management of GERD patients. In the study by Lee et al. [36], it was found that NERD (Non-erosive reflux disease) patients had higher HF power, reflecting increased parasympathetic activity, compared to patients with symptomatic erosive esophagitis (SE) and asymptomatic erosive esophagitis (AE). However, parameters related to autonomic tonus and LF/HF ratio did not differ significantly among the groups. Additionally, HRV analysis revealed lower autonomic tonus in patients with endoscopically confirmed esophagitis, even in the absence of symptoms, compared to NERD subjects, suggesting the influence of the structural state of the esophagus on autonomic nervous system (ANS) function, independent of symptomatology. The study by Chen et al. [37] demonstrated lower HF power in patients with erosive reflux disease (ERD) compared to NERD patients and healthy controls. Furthermore, NERD patients exhibited lower LF power and LF/HF ratio compared to ERD patients and controls. These findings suggest distinct autonomic function patterns between NERD and ERD patients, despite similar symptom severity. Importantly, the lack of significant correlation between HRV parameters and symptom severity scores implies that autonomic function and symptomatology may not be directly related in GERD patients. In our study, the control group exhibited lower values of HRV parameters (SD, TP, VL, HF, and SDNN) compared to the GERD group before treatment. However, there were no significant changes in HRV parameters before and after treatment in the GERD group, suggesting persistent ANS dysregulation (Table 3). These findings emphasize the role of ANS dysregulation in the pathophysiology of GERD and highlight the need for further research to elucidate its clinical implications.

The available literature on the sequencing changes of psychosomatic evaluations after treatment for GERD is limited. Jung et al. developed the SEQ-GERD questionnaire and cross-culturally validated the GERD-QOL questionnaire. They assessed the discriminative validity of SEQ-GERD by comparing changes in scores after four weeks of proton pump inhibitor administration [38]. Their study demonstrated good internal consistency and high test–retest reliability of SEQ-GERD. Scores on SEQ-GERD showed significant variations based on overall treatment effectiveness, with a significant decrease after drug treatment, supporting its discriminative validity. In our study, we observed notable linear reductions in GERDQ, GERDQLQ, RSI scores, and psychosomatic evaluations (BAI, BDI, and SSS-8) during the initial eight weeks of treatment. Additionally, our study contributes to the understanding of the relationship between stress and GERD pathophysiology. Stress has long been recognized as a contributing factor in exacerbating GERD symptoms. The stress response system, including the hypothalamic–pituitary–adrenal axis and the autonomic nervous system [39], can influence various physiological aspects of GERD, such as gastric motility, sphincter function, esophageal sensitivity, acid secretion, and esophageal clearance [40]. Our findings further support the link between stress and GERD by demonstrating changes in psychosomatic assessments and emphasizing the importance of stress management in the treatment of GERD. However, we observed that the improvements in psychosomatic status plateaued beyond the initial 2-month treatment period, suggesting the attainment of a stable level. Future research is needed to investigate the long-term trajectory of psychosomatic assessments in GERD patients and identify factors that may influence their changes over time.

A range of underlying mechanisms can contribute to persistent GERD symptoms, including poor compliance with prescribed PPI use, weakly acidic reflux (pH > 4), persistent regurgitation of large volumes of non-acid refluxate, and esophageal hypersensitivity [41]. Esophageal hypersensitivity is the main cause of refractory GERD in erosive esophagitis, often accompanied by a higher psychological burden. Recently, the concept of esophageal hypervigilance has emerged as a potential factor in refractory GERD. Esophageal hypervigilance is defined as the cognitive–affective process resulting from heightened awareness of discomfort and was introduced by Keefer et al. [42]. In their study, they analyzed 70 patients with PPI refractory symptoms using pH-impedance and psychometric tests. Interestingly, hypervigilance was found to be present in all phenotypes of GERD, regardless of symptom correlation, acid exposure, or normality. This suggests that the cognitive–affective process of hypervigilance may play a role in refractory GERD beyond the traditional categorizations based on symptomatology or acid exposure. To measure these cognitive–affective processes in refractory GERD, the seven-item Esophageal Hypervigilance and Anxiety Scale (EHAS-7) was recently developed, with a score greater than 13 indicating the need for psychological evaluation [43].

In our study, fifteen patients (15.3%) exhibited a refractory response, showing significantly higher baseline scores in GERDQ, GERDQLQ, RSI, BDI, and SSS-8 compared to the non-refractory group. Identifying clinical parameters that predict refractory GERD is crucial for developing personalized treatment strategies. Our findings revealed that a GERDQ score ≥ 10 and an SSS-8 score ≥ 12 were independent variables associated with refractory GERD. These findings support the idea that psychological distress and somatic symptoms are interconnected in refractory GERD patients. However, it is important to note that further research is needed to fully understand the underlying mechanisms and establish a definitive classification of refractory GERD as a somatic form disorder.

There were limitations in our study. The diagnosis of GERD solely was based on the presence of erosive esophagitis grades LA A and B endoscopically with typical symptoms, without comprehensive esophageal physiological testing for all patients. According to the Lyon Consensus [44], while these grades are commonly observed in patients with GERD, they are not considered sufficient for a definitive diagnosis. Further esophageal testing, such as pH-impedance monitoring and high-resolution manometry, are recommended, particularly for patients who are refractory to PPI treatment or when the diagnosis is uncertain. Our study did not include these additional diagnostic measures for the majority of participants, which may limit the generalizability of our findings. HRV was measured only at pre- and post-treatment phase. Although the lack of significant difference between the two HRV measurements suggests stability over time, a more comprehensive evaluation of HRV dynamics would provide a more thorough understanding of autonomic function in GERD. We did not control for caffeine intake and certain medications that could affect heart rate variability, potentially influencing results of HRV measurements. The follow-up period in our study was limited to 8 weeks, and the long-term trajectory of psychosomatic assessments in GERD patients remains unclear. Future studies with longer follow-up periods are needed to elucidate the long-term changes and factors influencing psychosomatic evaluations in GERD patients.

## 5. Conclusions

Our study contributes to the understanding of the relationship between psychosomatic evaluations, HRV, and GERD in patients with reflux esophagitis. We highlight the relevance of considering both physiological and psychological factors in GERD management. Our findings demonstrate the importance of comprehensive assessments in the treatment approach for GERD patients. Further research is needed to validate our results, explore underlying mechanisms, and investigate the long-term trajectory of psychosomatic evaluations in GERD patients.

## Figures and Tables

**Figure 1 life-13-01862-f001:**
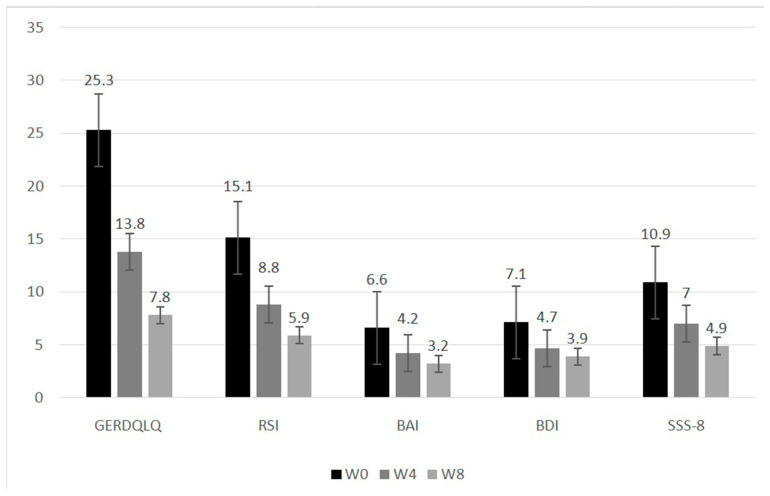
Sequential change of reflux symptom score and psychosomatic assessments after treatment. Abbreviations: GERDQLQ, GERD-Quality of Life Questionnaire; RSI, reflux symptom index; BAI, Beck anxiety inventory; BDI, Beck depression inventory; SSS-8, 8-item somatic symptom scale. All scores show a *p*-value of less than 0.001, indicating a statistically significant change in the linear trend.

**Figure 2 life-13-01862-f002:**
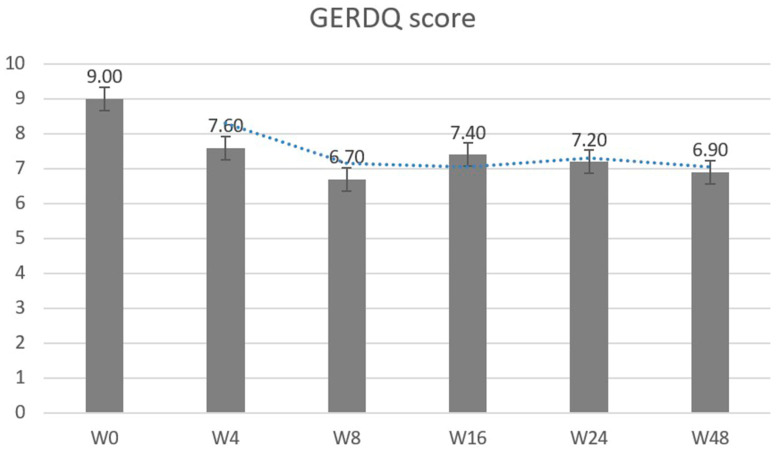
Sequential change of GERD-questionnaire over time. Abbreviations: GERDQ, GERD-questionnaire. The scores reveal a *p*-value of less than 0.001, and a statistically significant linear trend change between W0 and W8. The *p*-value is greater than 0.05 for the period from W8 to W48.

**Table 1 life-13-01862-t001:** Baseline characteristics between the GERD and control groups.

Variate	GERD Group (n = 98, %)	Control Group (n = 50, %)	*p*-Value
Age	50.6 ± 15.1	50.2 ± 13.1	0.854
Sex (M/F)	33/65 (33.7/66.3)	20/30 (40.0/60.0)	0.448
BMI	24.6 ± 4.1	24.5 ± 4.5	0.987
Smoking	9 (9.2)	5 (10.0)	0.872
Alcohol	13 (13.3)	2 (4.0)	0.090
Coffee	57 (58.2)	32 (64.0)	0.493
Tea	61 (62.2)	39 (78.0)	0.023
Betel nut	2 (2.0)	0 (0)	0.309
Spicy food	59 (60.2)	34 (68)	0.353
Sweaty food	67 (68.4)	40 (80)	0.135
Comorbidities			
Diabetes	15 (7.3)	4 (8)	0.209
Hypertension	25 (25.5)	5 (10)	0.080
Coronary artery disease	7 (7.1)	1 (2)	0.191
Cerebral vascular disease	0 (0)	0 (0)	-
Chronic kidney disease	1 (1.0)	0 (0)	0.474
Insomnia	55 (56.1)	14 (28.0)	0.001
GERD history	73 (74.5)	0 (0)	<0.001
Symptom score			
GERDQ	9.0 ± 2.4	5.9 ± 0.4	<0.001
GERDQLQ	25.3 ± 17.2	0	<0.001
RSI	15.1 ± 8.5	0.4 ± 0.2	<0.001
BAI	6.6 ± 6.6	0	<0.001
BDI	7.1 ± 6.7	0	<0.001
SSD-8	10.9 ± 6.8	0	<0.001

Abbreviations: GERDQ, GERD-questionnaire; GERDQLQ, GERD-Quality of Life Questionnaire; RSI, reflux symptom index; BAI, Beck anxiety inventory; BDI, Beck depression inventory; SSS-8, 8-item somatic symptom scale.

**Table 2 life-13-01862-t002:** Correlations between GERD symptoms and psychological scores.

	GERDQ (r and *p* Value)	GERDQLQ (r and *p* Value)	RSI (r and *p* Value)
W0			
BAI	r = 0.452, *p* < 0.001	r = 0.705, *p* < 0.001	r = 0.656, *p* < 0.001
BDI	r = 0.438, *p* < 0.001	r = 0.698, *p* < 0.001	r = 0.641, *p* < 0.001
SSS-8	r = 0.595, *p* < 0.001	r = 0.810, *p* < 0.001	r = 0.798, *p* < 0.001
W4			
BAI	r = 0.406, *p* < 0.001	r = 0.634, *p* < 0.001	r = 0.547, *p* < 0.001
BDI	r = 0.329, *p* = 0.001	r = 0.584, *p* < 0.001	r = 0.506, *p* < 0.001
SSS-8	r = 0.476, *p* < 0.001	r = 0.675, *p* < 0.001	r = 0.629, *p* < 0.001
W8			
BAI	r = 0.391, *p* < 0.001	r = 0.711, *p* < 0.001	r = 0.616, *p* < 0.001
BDI	r = 0.348, *p* < 0.001	r = 0.668, *p* < 0.001	r = 0.560, *p* < 0.001
SSS-8	r = 0.518, *p* < 0.001	r = 0.808, *p* < 0.001	r = 0.712, *p* < 0.001

Abbreviations: r, Pearson correlation coefficient; GERDQ, GERD-questionnaire; GERDQLQ, GERD-Quality of Life Questionnaire; RSI, reflux symptom index; BAI, Beck anxiety inventory; BDI, Beck depression inventory; SSS-8, 8-item somatic symptom scale.

**Table 3 life-13-01862-t003:** Heart rate variability parameters for the control and the GERD groups.

Variate/Time	Control Group	GERD Group W0	GERD Group W8	*p*-Value ^a^
HR	75.6 ± 10.1	73.3 ± 14.7	74.5 ± 12.2	0.394
SD	28.7 ± 12.6 *	39.8 ± 46.3	34.4 ± 31.0	0.319
RRIV (ms)	59.1 ± 100.1	58.1 ± 73.7	56.1 ± 79.4	0.857
TP (ms^2^)	784.0 ± 699.3 *	1556.2 ± 3324.4	1853.1 ± 5136.1	0.666
VLF (Hz)	331.7 ± 263.3 *	561.1 ± 954.5	468.5 ± 680.5	0.487
LF (Hz)	182.5 ± 201.4	396.0 ± 1121.8	349.3 ± 896.5	0.654
HF (Hz)	208.9 ± 318.2 *	618.5 ± 1644.6	443.4 ± 1209.6	0.319
NN	803.7 ± 113.1	840.2 ± 149.7	819.5 ± 130.0	0.122
ANSage (y/o)	55.3 ± 18.4	51.9 ± 21.5	53.0 ± 20.1	0.527
Bal	−0.43 ± 1.94	−0.75 ± 2.65	−0.44 ± 2.50	0.257
SDNN (ms)	28.3 ± 12.5 *	39.8 ± 46.3	34.2 ± 31.0	0.301
In (LF/HF)	0.077 ± 0.921	0.003 ± 1.106	0.126 ± 1.057	0.055
LF (%)	44.7 ± 18.6	40.8 ± 21.8	44.2 ± 21.6	0.155

Abbreviations: * reveals the significant difference between control group and W0 of GERD group; ^a^ the *p*-value between the W0 and W8 of GERD group; HRV, heart rate variability; HR, heart rate; SD, spectral distribution; RRIV, R-R interval variation; TP, total power; VLF, very low frequency; LF, low frequency; HF, high frequency; NN, Normal-to-Normal interval; ANS, autonomic nervous system; Bal, balance; SDNN, standard deviation of Normal-to-Normal.

**Table 4 life-13-01862-t004:** The predictors at baseline for refractory GERD at W8.

	Refractory GERD (n = 15)	Non-Refractory GERD (n = 83)	*p*-Value
Age	52.8 ± 15.0	50.2 ± 15.2	0.547
Sex(M/F) (n/%)	3/12 (20.0/80.0)	30/53 (31.6/63.9)	0.223
GERDQ	10.7 ± 1.8	8.7 ± 2.3	0.002
GERDQLQ	38.5 ± 12.5	22.9 ± 16.9	0.001
RSI	20.9 ± 7.8	14.1 ± 8.2	0.004
BAI	9.5 ± 6.5	6.1 ± 6.6	0.069
BDI	10.4 ± 5.3	6.5 ± 6.7	0.037
SSS-8	16.7 ± 7.2	9.8 ± 6.2	<0.001
HR	70.7 ± 13.5	74.5 ± 14.4	0.354
SD	30.5 ± 25.9	40.2 ± 45.4	0.426
RRIV (ms)	38.3 ± 35.2	64.2 ± 75.5	0.197
TP (ms^2^)	1194.6 ± 2239.4	1622.3 ± 3493.3	0.649
VLF (ms^2^)	500.6 ± 859.6	572.1 ± 975.3	0.791
LF (ms^2^)	224.3 ± 420.4	427.0 ± 1205.0	0.522
HF (ms^2^)	382.9 ± 956.9	591.0 ± 1595.6	0.626
NN	870.9 ± 159.3	826.0 ± 143.6	0.276
ANSage (y/o)	56.4 ± 21.0	51.1 ± 21.6	0.377
Bal	0.6 ± 3.4	−0.8 ± 2.4	0.054
SDNN (ms)	30.5 ± 25.9	40.1 ± 45.4	0.430
In (LF/HF)	0.3 ± 1.2	−0.1 ± 1.1	0.209
LF (%)	48.9 ± 23.3	41.2 ± 21.3	0.209

Abbreviations: GERDQ, GERD-questionnaire; GERDQLQ, GERD-Quality of Life Questionnaire; RSI, reflux symptom index; BAI, Beck anxiety inventory; BDI, Beck depression inventory; SSS-8, 8-item somatic symptom scale; HRV, heart rate variability; HR, heart rate; SD, spectral distribution; RRIV, R-R interval variation; TP, total power; VLF, very low frequency; LF, low frequency; HF, high frequency; NN, Normal-to-Normal interval; ANS, autonomic nervous system; Bal, balance; SDNN, standard deviation of Normal-to-Normal.

**Table 5 life-13-01862-t005:** Multivariate analyses of the factors predicting the refractory GERD.

Variants	Refractory GERD n = 15 (%)	Non-Refractory GERD n = 83 (%)	Multivariate OR (95% C.I.)	*p*-Value
GERDQ (≥10)	12 (80.0)	32 (38.6)	11.028 (2.147–56.654)	0.004
RSI (≥13)	12 (80.0)	46 (55.4)	1.350 (0.279–6.542)	0.709
BDI (≥17)	3 (20.0)	2 (2.4)	7.994 (0.736–86.879)	0.088
SSS-8 (≥12)	12 (80.0)	28 (33.7)	7.377 (1.636–33.256)	0.009

Abbreviations: GERDQ, GERD-questionnaire; RSI, reflux symptom index; BAI, Beck anxiety inventory; BDI, Beck depression inventory; SSS-8, 8-item somatic symptom scale.

## Data Availability

Not applicable.

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
