# Peer review of "Correlation between Psychosomatic Assessment, Heart Rate Variability, and Refractory GERD: A Prospective Study in Patients with Acid Reflux Esophagitis"

_life, 2023, doi:10.3390/life13091862_

Round 1
Reviewer 1 Report
This prospective case-control and cohort study aims to evaluate investigate the correlation between different psychosomatic evaluations and heart rate variability (HRV) among patients with reflux esophagitis and controls. The authors also analyzed the alterations in sequencing following PPIs treatment and identify potential predictive factors associated with refractory GERD. Generally speaking, this topic is important and the prospective design was solid with pre- and post-treatment comparison. The manuscript was well written and the conclusion was clear and practical. The reviewer only has a few comments and suggestions.
1. Please describe the inclusion and exclusion criteria of the control group. Are they healthy controls? Some of them have diabetes and CAD, which may affect the results of HRV and the reflux sensation. Or just did not have reflux symptoms (but may have other GI symptoms such as dyspepsia or abdominal pain) ? Did the controls undergo endoscopy to exclude EE?
2. All the patients had mild esophagitis, either Gr A or B. Did the authors specifically enroll the mild cases? How many of them with hiatal hernia?
3. It would be better to briefly describe the detail of GERDQ and RSI. The average score of GERDQ for the controls was 5.9, which may confuse the readers.
4. The HRV recording was only limited to 5 minutes which may be the cause of insignificant differences after PPI treatment. Please also describe the timing of HRV measurement.
5. It will be more informative to show the correlation between GERD symptom scores and the psychosomatic scores.
Minor
1. Table 1: Betel nut
2. Discussion. Last paragraph: I think it was 8 weeks rather than 48 weeks.
Reviewer 2 Report
Dear authors!
I read with interest your manuscript "Correlation between Psychosomatic Assessment, Heart Rate Variability, and Refractory GERD: A Prospective Study in Patients with Acid Reflux Esophagitis". The paper is based on the results of original single-center prospective controlled trial. Overall, the study is well-planned and the methods could allow to obtain (and reproduce, if necessary) the results. The manuscript evidently contains novelty and may be interesting to the readers. I have a few minor comments.
1. It seems that the title does not correspond with the study content and aim(s). The studied group was wider (subjects with reflux esophagitis and acid reflux symptoms); the presence of a control group is not mentioned; correlation coefficients are not provided in the text. Please, consider revision of the title to better reflect the content of your work and not to mislead the readers.
2. Please, explain the exclusion criteria (in context of the studied group): whether concomitant medications were taken into the account? This matter is important to confirm GERD, as some medications may provoke similar symptoms and even lead to the presence of esophagitis (like NSAIDs). According to the Rome criteria, time is also important factor to differentiate the condition between GERD and "non-GERD" - in GERD, acid-related symptoms should be present for more than 6 months, and should be actual for at least last 3 months before the diagnosis. According to the Lyon consensus, esophagitis of grade A or B is not sufficient to establish GERD. Thus, additional examinations should have ben performed to confirm the disease (i.e. MII-pH). It is mentioned in the paragraph 2.2 that "Refractory GERD patients were advised to undergo further esophageal manometry and 24-hour pH monitoring, as per clinical guidelines if patients agree it", but no data of these tests are not provided. If available, please, add the details of these studies, or explain how the condition of GERD was confirmed.
Who were the subjects of the control group and which criteria were used to enroll them to the study.
3. I wonder, whether conditions of HRV measurement were reproducible? Did you use special requirements (for example, did not allow to drink coffee or tee before the measurement, excluded intensive physical activity, the use of some medications that influence heart beat rate, provided some rest before tracing, etc?
4. Please, explain, how the measurements of HRV were performed twice in outpatient conditions (methods) but only one-time point (according to the discussion, section of limitations).
Minor comment: in the exclusion criteria, please, explain "alcohol misuse" - was it used to rub the skin or else? May the term "abuse" be better, instead?
The paper requires moderate language polishing
Reviewer 3 Report
The manuscript evaluating the correlation between GERD, HRV, and psychological symptoms is well-written and the conclusion is supported by the data. The study highlights that in GERD patients, psychological assessment should be done and along with HRV may be an indicator of treatment outcome. Please include statistics in all bar charts. Also, please check carefully for language as at a few places the words/sentences are either incomplete or missing, e.g., section 2.4.3-mirrors gastrointestinal...what, symptoms, reflux, pain, bleeding etc?,
Round 2
Reviewer 2 Report
Dear colleagues!
I've read your answers to the reviewer and the revised version of the manuscript. I suppose that you made a great job and the manuscript really turned to the better. However, I am still concerned about the matters covered in the Q2. I am pretty sure that you are aware of all current guidelines, and I am not to discuss them again. As there is no way to confirm the relationship of the refractory flow of the disease with gastroesophageal reflux in your patients at the moment, please, add the information about the Lyon approach to RE grades A&B, and lack of the possibility to clearly measure association of the studied condition with reflux except 2 cases that underwent MII-pH studies.
NA
